# Assimilation of SMAP Products for Improving Streamflow Simulations over Tropical Climate Region—Is Spatial Information More Important than Temporal Information?

**Manh-Hung Le** [1,*], **Binh Quang Nguyen** [2], **Hung T. Pham** [2], **Amol Patil** [3], **Hong Xuan Do** [4], **RAAJ Ramsankaran** [5], **John D. Bolten** [6] and **Venkataraman Lakshmi** [1]

1 Department of Engineering Systems and Environment, University of Virginia, Charlottesville, VA 22904, USA; vl9tn@virginia.edu
2 Faculty of Water Resources Engineering, The University of Danang—University of Science and Technology, Da Nang 550000, Vietnam; nqbinh@dut.udn.vn (B.Q.N.); pthung@dut.udn.vn (H.T.P.)
3 Chair of Regional Climate and Hydrology, Institute of Geography, University of Augsburg, 86159 Augsburg, Germany; amol.patil@geo.uni-augsburg.de
4 Faculty of Environment and Natural Resources, Nong Lam University, Ho Chi Minh City 700000, Vietnam; doxuanhong@hcmuaf.edu.vn
5 Hydro-Remote Sensing Applications (H-RSA) Group, Department of Civil Engineering, Indian Institute of Technology, Mumbai 400076, Maharashtra, India; ramsankaran@civil.iitb.ac.in
6 Hydrological Sciences Lab, NASA Goddard Space Flight Center, Greenbelt, MD 20771, USA; john.bolten@nasa.gov
* Correspondence: hml5rn@virginia.edu

**Abstract:** Streamflow is one of the key variables in the hydrological cycle. Simulation and forecasting of streamflow are challenging tasks for hydrologists, especially in sparsely gauged areas. Coarse spatial resolution remote sensing soil moisture products (equal to or larger than 9 km) are often assimilated into hydrological models to improve streamflow simulation in large catchments. This study uses the Ensemble Kalman Filter (EnKF) technique to assimilate SMAP soil moisture products at the coarse spatial resolution of 9 km (SMAP 9 km), and downscaled SMAP soil moisture product at the higher spatial resolution of 1 km (SMAP 1 km), into the Soil and Water Assessment Tool (SWAT) to investigate the usefulness of different spatial and temporal resolutions of remotely sensed soil moisture products in streamflow simulation and forecasting. The experiment was set up for eight catchments across the tropical climate of Vietnam, with varying catchment areas from 267 to 6430 km$^2$ during the period 2017–2019. We comprehensively evaluated the EnKF-based SWAT model in simulating streamflow at low, average, and high flow. Our results indicated that high-spatial resolution of downscaled SMAP 1 km is more beneficial in the data assimilation framework in aiding the accuracy of streamflow simulation, as compared to that of SMAP 9 km, especially for the small catchments. Our analysis on the impact of observation resolution also indicates that the improvement in the streamflow simulation with data assimilation is more significant at catchments where downscaled SMAP 1 km has fewer missing observations. This study is helpful for adding more understanding of performances of soil moisture data assimilation based hydrological modelling over the tropical climate region, and exhibits the potential use of remote sensing data assimilation in hydrology.

**Keywords:** soil moisture; Vietnam; SWAT; Ensemble Kalman Filter; small catchments

## 1. Introduction

In recent years, soil moisture (SM) has been increasingly investigated in hydrological research as it has a strong influence on the interaction between different components within the hydrological cycle [1–3]. The SM content is a key variable that controls most of the land surface hydrological processes and thus is considered one of the most important parameters

in land surface hydrology models [4]. The increased need for satellite-based soil moisture information has led to the launch of many satellite missions to provide more accurate SM estimates at the global scale [5,6] that could be used to substitute in-situ SM observations that only cover a very limited portion of the land surface [7]. These SM products include ASCAT (Advanced SCATterometer) [8], SMOS (Soil Moisture and Ocean Salinity) [9], AMSR-E (Advanced Microwave Scanning Radiometer for the Earth Observing System onboard the Aqua satellite) [10], AMSR-2 (Advanced Microwave Scanning Radiometer 2 onboard the Global Change Observation Mission—Water satellite) [11] and SMAP (Soil Moisture Active Passive) [12]. All of these SM data products are freely accessible, providing an opportunity to integrate SM information into hydrological models across the globe.

Owing to the release of the above-mentioned data products, assimilation of soil moisture (SM) in hydrological simulations has received much attention within the past decade. Specifically, of 150 studies conducted during the period of 2001–2021 on soil moisture assimilation in hydrology modelling, nearly ninety percent have been published since 2012 (see Supplementary Figure S1). A number of studies have assessed remotely-sensed SM assimilation in various hydrological applications, including flood prediction [13,14], water balance estimation [15], and streamflow forecast [16,17], along with agricultural monitoring and forecasting [18,19]. These studies have established a new frontier in hydrological research to take advantage of SM estimates from space to inform hydrological modeling.

However, satellite-based SM products also have several limitations, including shallow penetration depth (typically shallower than or equal to 5 cm) and relatively coarse spatial resolutions (larger than or equal to 9 km) [12]. Therefore, the SM observed from space may often improve the top-soil layer estimation, unless carefully integrated into a soil moisture or hydrologic model through direct insertion or data assimilation. Although several studies [20] have shown that coarse spatial resolutions of remote sensing soil moisture could be useful in improving streamflow simulations, many studies have pointed out the limitations of low spatial resolutions of soil moisture in data assimilation, especially in small catchments [21] or in flash flood forecasting [22].

To overcome the low spatial resolution of satellite-based SM products, several studies have proposed different downscaled algorithms to obtain a finer soil moisture dataset in space. These algorithms can be classified into three primary types, including (i) methods based on a satellite data combination of high and low resolution satellite data using active sensors [23,24], and visible, infrared and thermal sensors [25–28]; (ii) methods based on the relationship between SM and other geophysical variables that exist at a finer spatial resolution [29,30]; (iii) methods based on mathematical modelling (e.g., land surface modelling) to simulate coarse resolution remotely sensed SM to a fine resolution model to update SM outputs [31,32].

On the other hand, compared to native resolution satellite-based products, downscaled satellite-based SM products are prone to having shorter data records, complicating typical data assimilation methodologies. For instance, with the first downscaling method mentioned above, a widely-used algorithm is a thermal inertia principle-based algorithm [33]. This algorithm utilizes the universal relationship between land surface temperature (LST), vegetation index, soil wetness, and evapotranspiration to quantify SM as a function of LST and normalized different vegetation index (NDVI). However, the LST dataset, which is often retrieved from earth observations, often has large spatial and temporal gaps, resulting from atmospheric conditions (e.g., cloud and cloud shadows) [34]. Consequently, these LST's gaps will cause gaps in space and time for downscaled SM product and result in an absence of temporal time series during the data assimilation process. Although efforts exist to fill the gaps from LST before the downscaling step [33,35], the challenge of supplementing temporally-downscaled SM data for assimilation still remains.

Investigation of the trade-offs between temporal and spatial resolution of remotely sensed SM products for constraining hydrologic models is an area of research that requires more attention. In a study of two catchments in Central Italy, Azimi et al., 2020 [36] examined the benefit of having more frequent SM observations (temporal timescale) in

streamflow simulation. The authors concluded that reduced temporal sampling from a remotely sensed soil moisture product could significantly reduce model performance, indicating that temporal resolution likely plays a more important role than spatial resolution in constraining the model. On the other hand, a study using SMAP soil moisture data assimilation in a community-based hydrologic model indicates that downscaled SMAP 1 km would improve the accuracy of streamflow simulation (normal streamflow conditions), rather than the model using coarse resolution SMAP 9 km data [13].

In addition, the impact of the number, size, and nature of the hydrologic catchment requires further investigation—few studies have addressed the potential impacts of catchment characteristics on SM-based DA schemes. A majority of studies have examined the DA schemes in a focused area, and typically over relatively few catchments (e.g., <4), making it difficult to make conclusive statements on the utility of such DA approaches (see Table 1 and Supplementary Figure S2). Several studies that have included large samples of catchments concluded that a hydrological model with a SM-based DA framework may not significantly improve streamflow simulations, compared to the hydrological model without the DA [37,38].

Model complexity, and heterogeneous land surface characterization and meteorological forcing, can result in varying levels of uncertainty and model accuracy, issues not easily corrected through data assimilation. In fact, DA-driven hydrologic models often exhibit mixed results across climatic conditions. This is an active area of research, and more studies are encouraged. Currently, most studies focus on temperate regions (see Table 1). In the tropical climate, streamflow is often of great variation, due to the impacts of large-scale phenomena such as ENSO on the seasonal and year-to-year variation in soil moisture, which results from the high variability in rainfall [39]. Any technique such as DA that could enhance hydrological model performances in the tropical climate region is essential, but such studies have rarely been investigated [40], owing to the difficulty of accessing streamflow records over these regions [41].

Here, we build off of these previous studies and attempt to demonstrate the utility of satellite-based soil moisture for streamflow simulation, as well as assessing the impacts of temporal and spatial resolution on the model accuracy. We carefully investigate the application of two remotely sensed SM products (SMAP 9 km and downscaled SMAP 1 km) to examine whether spatial–temporal resolution has a substantial impact on the performance of the hydrological model to simulate streamflow through a data assimilation (DA) framework. We carried out the experiment over eight catchments across Vietnam—a tropical country that is under-represented in the literature. The hydrological Soil and Water Assessment Tool (SWAT) model [42] is selected as it performs well in numerous studies in the studied region [43–47], and there are several studies that have successfully implemented the DA framework in the SWAT model [36,48]. We selected the Ensemble Kalman Filter (EnKF) [49] as the DA algorithm due to its popularity in many hydrological assimilation works [31,38,50].

Section 2 presents eight catchments together with the selected datasets while Section 3 provides a brief description of the hydrological SWAT model and data assimilation scheme that were used to conduct this study. Section 4 provides a comprehensive assessment of the findings, focusing on the discrepancies of model performance under different DA schemes. Section 5 concluded the study findings.

**Table 1.** Summary of selected studies on remote sensing soil moisture data assimilation in hydrologic models. These studies were investigated in terms of climate region, number of studied catchments, used remotely sensed (RS) soil moisture (SM) datasets, data assimilation (DA) technique with hydrologic models. More details on recent studies (2015-present) can be found in Supplementary Material Figure S2.

| References | Climate Region | Catchments/RS SM Datasets | DA[(*)]/Hydrological Models [(**)] | Main Findings |
|---|---|---|---|---|
| Jadidoleslam et al., 2021 [37] | Cold | 131/SMAP, SMOS | EnKF, EnKFV/HLM | DA driven models reduce the peak error and could be useful for the application of satellite soil moisture for operational real-time streamflow forecasting. |
| Abbaszadeh et al., 2020 [13] | Temperate | 4/SMAP | EPFM/WRF-Hydro | Assimilation of SM could improve streamflow simulation during flooding from hurricane Harvey in 2017, with a promising result from SM at 1 km. |
| Baguis et al., 2017 [51] | Temperate | 1/ASCAT | EnKF/SCHEME | The DA algorithm could be a diagnostic tool to detect weakness in a model and to improve its performance. |
| Patil and Ramsankaran, 2018 [14] | Temperate | 2/SMOS, ASCAT | EnKF/SWAT | A coupling Soil Moisture Analytical Relationship with EnKF could successfully update the sub-surface SM and streamflow components simulation. |
| Laiolo et al., 2016 [20] | Temperate | 1/EUMET-SAT H-SAF, SMOS | Nudging/Continuum | Streamflow prediction for a small basin using a distributed hydrological model could be improved with the assimilation of soil moisture estimated from coarse spatial resolution remotely sensed products. |
| Behera et al., 2019 [15] | Tropical | 1/AMSR-E | Kalman Filter/VIC | DA driven models could improve soil moisture in root zone and water balance estimation. |
| Azimi et al., 2020 [36] | Temperate | 2/SMAP, SACAT, CATSAR-SWI | EnKF/SWAT | Both active and passive-based SM driven simulation generally improved streamflow simulation. The impact of frequency of soil moisture observation on data assimilation performances in small catchments was discussed. |
| Lü et al., 2016 [52] | Arid | 2/ASCAT | EnKF/HBV | A combined surface soil moisture and snow depth data assimilation into a hydrological model was proposed to improve streamflow estimation in cold and warm season headwater watersheds. |
| Yang et al., 2021 [31] | Temperate | 3/ESA CCI, SMAP | EnKF/DDRM | Assimilation of soil moisture products in high spatial gridded modelling could increase DA performances in terms of simulating profile soil moisture. |
| De Santis et al., 2021 [38] | Cold, Temperate | 775/ESA CCI | EnKF/MISDc-2L | An assessment of large-scale DA experiments in hydrological model streamflow simulation was carried out over Europe. This study also considered impacts of vegetation density, topographical complexity and basin area on the DA performances. |
| Loizu et al., 2018 [53] | Temperate | 2/ASCAT | EnKF/MISDc, TOPLATS | This study examined the impacts of three different re-scaling techniques on SM data assimilation for two hydrological models. A careful evaluation for observation error and re-scaling technique is recommended for successful implementation of a data assimilation framework. |

Note: (*) Acronyms for data assimilation techniques: 'EnKF' Ensemble Kalman Filter, 'EnKFV' EnKF include time-varying error variances, 'EPFM' Evolutionary Particle Filter with Markov Chain Monte Carlo. [(**)] Acronyms for hydrologic models: 'HLM' Hillslope Link Model, 'WRF-Hydro' Weather Research and Forecasting Hydrological model, 'SCHEME' SCHEldt-MEuse, from the names of the two major rivers of Belgium, 'SWAT' Soil and Water Assessment Tool, 'VIC' Variable Infiltration Capacity, 'HBV' Hydrologiska Byråns Vattenbalansavdelning, 'DDRM' Digital Elevation Model (DEM) based distributed rainfall-runoff model, 'MISDc-2L' Modello Idrologico Semi-Distribuito in continuo-2 layers, 'TOPLATS' TOPMODEL-Based Land Surface-Atmosphere Transfer Scheme.

## 2. Materials and Methods

### 2.1. Catchment Sites and Its Streamflow Observations

We collected daily 2013–2019 streamflow time series from eight hydrological stations across Vietnam with their characteristics presented in Table 2. The in-situ streamflow datasets have been used to calibrate the hydrological models for each catchment, and evaluate the performance of hydrological simulations with and without DA. These catchments were selected based on several study objectives. Firstly, they have a variety of catchment sizes so that we could examine the impacts of the spatial resolution of SMAP products on the data assimilation algorithm. Secondly, they are in contrasting climate conditions and geographic coordinates. Therefore, they have different runoff regimes and soil moisture patterns (Figure 1), which are useful for drawing a general conclusion on our experiment. Lastly, all catchments have passed homogeneity time series testing, and have natural runoff conditions due to the lack of manmade structures (i.e., weirs, dams, etc.). These conditions enable us to isolate the impact of the DA methods by removing potential changes in streamflow dynamics due to human activities. Details on testing of homogeneity time series and checking of natural catchment conditions can be found in Do et al., 2022 [54].

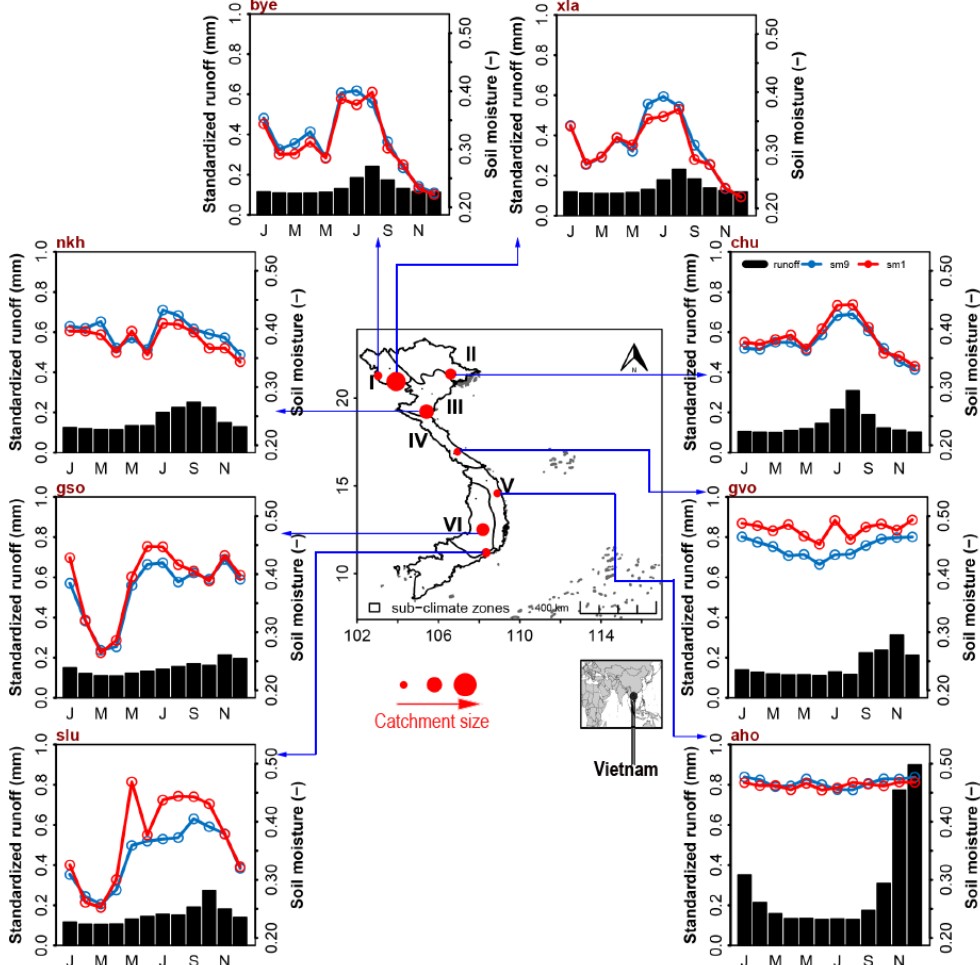

**Figure 1.** Locations of eight catchments (red circle represents catchment centroid) in Vietnam, and their monthly averaged runoff (black bar), monthly averaged soil moisture estimated from SMAP 9 km (SM9, blue line), and monthly averaged soil moisture estimated from SMAP 1 km (SM1, red line). The runoff values were calculated based on the period of 2013–2019, while soil moisture values (volume soil moisture) were calculated based on the period of 2017–2019. A rescaling has been applied for the runoff time series to compare its variation across catchments. The circle size indicates relative size of the catchment. The Roman numerals indicate contrasting climate regions where the studied catchments located in. These regions are defined following [55].

**Table 2.** Description of hydrological stations used in this study. Average runoff characteristics in each catchment (min, median, mean, max) are based on time series 2013–2019. NDVI is the average NDVI value for each catchment during 2017–2019 extracted from MODIS MOD13Q1 250 m product. SM9 and SM1 stand for the percentage of available SMAP 9 km and downscaled SMAP 1 km during the data assimilation period (2017–2019), respectively.

| Full Name | Short Name | Long. (Degree) | Lat. (Degree) | Area (km$^2$) | Min (mm/d) | Median (mm/d) | Mean (mm/d) | Max (mm/d) | NDVI (-) | SM9 (%) | SM1 (%) |
|---|---|---|---|---|---|---|---|---|---|---|---|
| Giavong | gvo | 106.93 | 16.93 | 267 | 0.09 | 0.91 | 2.49 | 136.56 | 0.801 | 42.37 | 9.68 |
| Anhoa | aho | 108.90 | 14.57 | 383 | 0.36 | 1.87 | 7.54 | 254.91 | 0.628 | 31.78 | 10.41 |
| Banyen | bye | 103.03 | 21.27 | 638 | 0.21 | 0.65 | 1.51 | 33.04 | 0.740 | 42.56 | 21.46 |
| Songluy | slu | 108.34 | 11.19 | 964 | 0.04 | 0.51 | 2.02 | 42.30 | 0.808 | 41.74 | 5.84 |
| Chu | chu | 106.60 | 21.37 | 2090 | 0.02 | 0.25 | 1.79 | 99.22 | 0.736 | 31.78 | 12.24 |
| Giangson | gso | 108.19 | 12.51 | 3100 | 0.18 | 1.28 | 1.95 | 28.71 | 0.753 | 31.78 | 11.6 |
| Nghiakhanh | nkh | 105.41 | 19.22 | 4024 | 0.32 | 1.16 | 2.39 | 92.11 | 0.770 | 31.78 | 14.52 |
| Xala | xla | 103.92 | 20.94 | 6430 | 0.13 | 0.89 | 1.64 | 24.72 | 0.686 | 34.16 | 16.62 |

*2.2. Climatic Datasets*

The climatic datasets forced into the hydrological model in this study are daily precipitation from GPM IMERG and daily maximum and minimum air temperature from NCEP CFSR V2. A detailed description of these datasets is given below.

2.2.1. GPM IMERG Precipitation

The half-hour 0.1 degree GPM IMERG Final run V6 (hereafter IMERG) [56] was downloaded from NASA Goddard Earth Science Data and Information Services Center (GES DISC, https://disc.gsfc.nasa.gov/, accessed on 28 January 2022). Daily precipitation totals were calculated by summing 24-h periods beginning at 19:00 UTC the day prior to the day of the record to match with the local daily rainfall collection time frame. Satellite precipitation has been shown to favorably compare with rain gages in various locations [57–59].

2.2.2. NCEP CFSR V2 Air Temperature

The 6-hour CFSR V2 for maximum and minimum air temperature [60] was downloaded from the National Center for Atmospheric Research (NCAR, https://rda.ucar.edu/, accessed on 28 January 2022) Data Archive. Depending on the parameters, the available resolution varies from 0.3 degrees to 2.5 degrees. In this study, we selected the finest resolution of 0.3 degrees. We obtained the maximum and minimum air temperature every 6 h, and selected the maximum and minimum among these four periods per day to estimate the daily maximum and minimum air temperature, respectively.

*2.3. Remotely Sensed Soil Moisture Datasets*

We obtained two soil moisture (SM) products originating from Soil Moisture Active Passive (SMAP). These products have exhibited their potential use in water resources and hydrology in the studied region [61,62], and are the data assimilation variables (i.e., state variables) which serve as the observed soil moisture to assimilate into the hydrological model.

2.3.1. Soil Moisture Active Passive

The 9 km SMAP Level-3 (hereafter SM9) was obtained from the National Snow and Ice Data Center (NSIDC DAAC, http://nsidc.org/data/smap, accessed on 28 January 2022). The SMAP provides, at approximately 06:00 and 18:00 local time (LT), soil moisture data in descending and ascending orbits, respectively. In this study, to match with daily simulation time in the study region, the SMAP ascending overpass time (18:00 LT) is selected as the observed soil moisture for a day. The accuracy for the SMAP data is designed with μRMSE of 0.04 m$^3$/m$^3$ [5].

2.3.2. Downscaled Soil Moisture Active Passive

Based on the assumption that daily soil moisture was negatively associated with the change in daily temperature under varying vegetation conditions, Fang et al., 2018 [63]; Fang et al., 2020 [27] proposed a linear regression model to estimate the daily soil moisture condition with known daily temperature and vegetation index. Using this linear regression model, we can create a finer spatial resolution for SM from high spatial resolutions of land surface temperature (reflecting the change in daily temperature) and of NDVI (reflecting the vegetation conditions). In this way, very high spatial soil moisture from SMAP— downscaled SMAP—has increased from 9-km to 1-km resolution (hereafter SM1). This SM1 product has been validated in CONUS [27], Australia [64], and at a global scale [33]. In this study, we obtained SM1 from the global scale product [33], and extracted the 18:00 LT, similar to the SM9.

## 3. Methodology

### 3.1. Principle of the Hydrological SWAT Model in Streamflow Simulation

The Soil and Water Assessment Tool (SWAT) is a physically based, semi-distributed hydrologic model that simulates various hydrologic variables at time steps (i.e., daily, monthly, and yearly) at catchment scale. The Hydrologic Response Unit (HRU) is the basic spatial unit of the SWAT model. Runoff generation is estimated at the HRU level, and is then routed to sub-basins and, subsequently, to the entire basin [65]. In the SWAT model, runoff generation is the sum of three components—surface runoff ($Q_{surf}$), lateral flow ($Q_{lat}$) and groundwater ($Q_{gw}$). The mathematical expression of these three components is described in the following.

The surface runoff process is a function of daily rainfall ($R_{day}$, unit in mm) and the retention parameter ($S$, unit in mm) based on the empirical formula using Soil Conservation Service (SCS) Curve Number (CN) method (SCS, 1972).

$$Q_{surf} = \frac{\left(R_{day} - 0.2{\cdot}S\right)^2}{R_{day} + 0.8{\cdot}S} \tag{1}$$

The retention parameter $S$ is calculated as follows.

$$S = S_{max}\left(1 - \frac{SW}{SW + \exp(w_1 - w_2{\cdot}SW)}\right) \tag{2}$$

where $S_{max}$ is the maximum value the retention parameter can obtain from any given day (mm). $SW$ is the total soil moisture (in mm) of the entire profile excluding the amount of water held at the wilting point. $w_1$ and $w_2$ are shape coefficients.

The shape coefficients ($w_1$ and $w_2$) are calculated as follows:

$$w_1 = ln\left[\frac{FC}{1 - S_3{\cdot}S_{max}^{-1}} - FC\right] + w_2{\cdot}FC \tag{3}$$

$$w_2 = \frac{\left(ln\left[\frac{FC}{1 - S_3{\cdot}S_{max}^{-1}} - FC\right] - ln\left[\frac{SAT}{1 - 2.54{\cdot}S_{max}^{-1}} - SAT\right]\right)}{(SAT - FC)} \tag{4}$$

where $FC$ is field capacity, $SAT$ is the amount of water when the soil profile is completely saturated (mm), and 2.54 is the retention parameter at the $CN = 99$. $S_3$ (mm) and $S_{max}$ (mm) are retention parameters, calculated given $CN_1$ (dry condition) and $CN_3$ (normal condition) as follows.

$$S = 25.4{\cdot}\left(\frac{1000}{CN} - 10\right) \tag{5}$$

where $S_{max} = 25.4{\cdot}\left(\frac{1000}{CN_1} - 10\right)$, and $S_3 = 25.4{\cdot}\left(\frac{1000}{CN_3} - 10\right)$

The $CN_1$ and $CN_3$ are calculated given $CN_2$ value (given as SWAT model input) as follows:

$$CN_1 = CN_2 - \frac{20 \cdot (100 - CN_2)}{(100 - CN_2 + \exp[2.533 - 0.0636 \cdot (100 - CN_2)])} \tag{6}$$

$$CN_3 = CN_2 \cdot \exp[0.00673 \cdot (100 - CN_2)] \tag{7}$$

After the surface runoff is formed, the rest of water infiltrates the land to generate soil water inflow. Lateral flow ($Q_{lat}$, unit in mm) in each soil layer is given as follows:

$$Q_{lat} = 0.024 \cdot \left( \frac{2 \cdot SW_{ly.excess} \cdot K_{sat.ly} \cdot slp}{\varphi_d \cdot L_{hill}} \right) \tag{8}$$

where $K_{sat.ly}$ is saturated hydraulic conductivity (mm/h) at layer $i$ ($i$ = 1, 2, 3), $slp$ is the steepness of a slope (m/m), $\varphi_d$ is the drainable porosity of the soil layer (mm/mm), and $L_{hill}$ is the hillslope length (m). In addition, $SW_{ly.excess}$ is the amount of soil water that exceeds field capacity at layer $i$ ($i$ = 1, 2, 3), is given as follows.

$$SW_{ly, excess} = SW_{ly} - FC_{ly} \; if \; SW_{ly} > FC_{ly}$$
$$SW_{ly,excess} = 0 \; if \; SW_{ly} \leq FC_{ly} \tag{9}$$

where $SW_{ly}$ and $FC_{ly}$ are the water content of the soil layer $i$ ($i$ = 1, 2, 3), on a given day (mm) and at field capacity (mm).

The $SW_{ly}$, if it exists, also generates deep percolation ($Q_{perc, ly}$, unit in mm) (from one layer to the underlying layer) as follows:

$$Q_{perc,ly} = SW_{ly,excess} \left( 1 - \exp \frac{-\Delta t \cdot K_{sat,ly}}{SAT_{ly} - FC_{ly}} \right) \tag{10}$$

where $t$ is the time step (hour). The soil water at the third layer percolates to vadose zones and groundwater (shallow aquifer layer). We focus on assimilating the soil moisture dynamic but do not consider the 'revap' process—water may move from shallow aquifers to overlaying unsaturated zones.

### 3.2. Setup the Hydrological SWAT Model

To set up the SWAT model across various catchment size basins, we (i) defined the same threshold to create a river network (i.e., 30 km$^2$) when using the DEM to delineate watersheds; (ii) set up a similar slope band setup (0-, 5-, 10-, 30-, and 50-degree).

For the climatic data inputs, using Thiessen polygon areal weighted average method [66], we calculated the mean areal precipitation for each sub-basin from gridded IMERG precipitation and the mean areal air temperature (i.e., maximum and minimum) for each sub-basin from gridded CFSR V2. Therefore, the precipitation and air temperature points as input for the SWAT models are equal to the total of the sub-basins.

To create HRU units, DEM, land use, and soil data are required. The 90-m void-filled digital elevation model (DEM) has been obtained from the hydrological data and maps based on SHuttle Elevation Derivatives at multiple Scales (HydroSHEDS, hydrosheds.org) [67,68]. The HydroSHEDS DEM has provided a reliable watershed delineation for the given studied basins with the difference between the catchment area generated from HydroSHEDS DEM and metadata being within ±15%. The 500-m land use land cover presented in this study is obtained from Collection 6 MODIS Land Cover (MCD12Q1 and MCD12C1) [69] from the Land Processes Distributed Active Archive Center (LP DAAC, https://lpdaac.usgs.gov/products/mcd12q1v006/, accessed on 28 January 2022). The MODIS Land cover provides 17 different land cover types annually from 2001 to 2019. This study obtained 2016 land cover as representing the land use in the given studied areas. Furthermore, this study reclassified the original 17 land cover types to 10 land cover types to match

with the SWAT format. This study used 1-km Harmonized World Soil Database (HWSD) version 1.2 maintained by the Food Agriculture Organization (FAO, http://www.fao.org, accessed on 28 January 2022) [70,71]. To prepare soil inputs for SWAT, we reclassified the HWSD's soil mapping unit (SMU) to the FAO soil symbol, assigned soil properties for each soil layer using the HWSD database, and used soil water characteristics equations from Saxton and Rawls (2006) to create a proper user soil format for SWAT. Normally, two soil layers' profiles are created (i.e., 0–300 mm, 300–1000 mm). However, SMAP can only measure soil moisture at the depth of 0–50 mm. Therefore, to have a realistic assimilation process, we re-classified the soil profile of SWAT from two layers to three layers (0–50 mm, 50–300 mm, and 300–1000 m) [16]. All described spatial processing (watershed delineation and HRU creation) have been conducted in QGIS v2.6.1 and QSWAT v1.7 [72]. Summarized descriptions of previously described datasets in Section 2 and DEM, soil, land use datasets for setup SWAT model are given in Table 3. The detailed climatic conditions, catchment attributes and model setup information (sub-basins and HRUs) are provided in the Table A1.

**Table 3.** Description of data used for SWAT and data assimilation framework in this study.

| Attributes | Data Type | Description | Period(s)/Resolution | Sources |
|---|---|---|---|---|
| Climatic data | Precipitation | IMERG Final Run V6 | 2011–2019/0.10° | [56] |
| | Max-, min- air temperature | CFSR vs2 | 2011–2019/0.25° | [60] |
| Catchment attributes | Land use land cover | MCD12Q1 | 2016/500 m | [69] |
| | Soil | HWSD | -/1 km | [70] |
| | Digital Elevation Model | HydroSHEDS | -/90 m (3 s) | [67] |
| Data assimilation variable | Soil moisture | SMAP | 2015–2019/9-km | [12] |
| | Soil moisture | Downscaled SMAP | 2015–2019/1-km | [33] |
| Ground data | Streamflow | Eight hydrological stations | 2013–2019 | VMHA * |

* VMHA Vietnam Meteorological and Hydrological Administration.

With respect to the parameterization of the SWAT model, we selected the warm-up, calibration and validation periods as 2011–2012, 2013–2016, and 2017–2019, respectively. Thirteen different parameters (see Table A2), which impact surface runoff, evaporation, soil moisture, and channel routing in the SWAT model, have been chosen for the parameterization. The parameters' turning process was undertaken with the SUFI-2 algorithm that is built in to the SWAT-CUP software [73]. In the end, we optimized the best suitable parameters for each catchment for daily streamflow simulation. The SWAT driven simulation at this step is considered as a deterministic SWAT model.

### 3.3. Data Assimilation—Ensemble Kalman Filter (EnKF)

3.3.1. Bias Correction of Observed SM and Ensembles Generation

The EnKF is a sequential data assimilation technique that is best applied using unbiased observations. To limit error covariance of the modeled and observed states in the EnKF, systematic errors between satellite SM retrievals and model states must be corrected before assimilation. It is assumed that long-term statistics of model states are consistent with those of in-situ SM [74], thus the model simulated states are normally used to correct biases in the satellite SM retrievals. We first estimated observed SM (from SM9 and SM1) for the topsoil layer (0–50 mm) for each HRU by calculating average satellite-observed SM at each sub-basin using the areal weighted average method [66]. The systematic differences between modeled (i.e., open loop) and remote sensing of soil moisture were then corrected using a mean-variance approach [16]. From the mean-variance matching, both model simulated SM and observed SM were estimated on monthly timescale and HRU spatial scale. The bias corrected SM was then used for the next analysis.

We generated 100 ensembles using the Latin Hypercube sampling technique [16] and defined ranges of error variances used for generating ensemble of model forcing, soil field capacity and observed soil moisture states (see Table A3). Since we employed

this EnKF data assimilation framework in multiple catchments with different climatic conditions, as well as with two different SM products, we assessed the error variances for each perturbed variable.

### 3.3.2. EnKF Algorithm

The EnKF is a Monte Carlo approximation (i.e., ensemble) of the standard Kalman Filter for use in a non-linear model. It uses an ensemble of modelled states in a Bayesian-based auto-recursive analysis framework to optimally merge model estimates with state observations (i.e., SM). The EnKF was operated in two steps as follows.

Step 1—Uncertainties from the ensemble of modeled forecasts and ensemble of observations.

During the soil water routing progress at any time step, at each HRU, the ensemble of model state (i.e., soil moisture) forecast is given as below.

$$x_{k+1}^{i-} = M\left(x_k^{i+}, U_k^i\right) + w_{k+1} \tag{11}$$

where $M$ is a non-linear model, which is the hydrological SWAT model in this study. The superscript $i$ represents a matrix of state ensembles with the forecast state (sign '-'), and analyzed state (sign '+'). The subscript $k$ represents the time step. $U_k^i$ is an ensemble of the model forcing. In this case, $U$ is perturbed precipitation. $w_{k+1}$ is Gaussian white noise representing the error due to uncertainties of forcing and model structure. Further, the ensemble of observations using the ensemble of states is calculated as follows.

$$\hat{z}_{k+1}^i = H_k x_{k+1}^{i-} + v_{k+1} \tag{12}$$

where $\hat{z}$ is the model predicted observation ensemble at time $k + 1$. $H$ is the observation operation to match the model states with the observations. Here, $H$ is the areal weighted average soil moisture at HRU. $v$ is the observation error, with separation of model errors and assumption of normally distributed with covariance $\sum_{k+1}^z$.

Step 2—Data assimilation progress.

The model forecasts are updated towards observations using Kalman Gain matrix ($K$)'s weights as,

$$x_{k+1}^{i+} = x_{k+1}^{i-} + K\left(z_{k+1}^i - \hat{z}_{k+1}^i\right) \tag{13}$$

where $x_{k+1}^{i-}$, $x_{k+1}^{i+}$ represent an ensemble of model forecasts and of state after assimilation, respectively. $z_{k+1}^i$ is an observation ensemble generated using the observation covariance matrix $\sum_{k+1}^z$.

The best linear unbiased estimation of $x_{k+1}^{i+}$ when the Kalma gain is calculated as,

$$K = \sum_{k+1}^{XZ}\left[\sum_{k+1}^{ZZ} + \sum_{k+1}^{Z}\right]^{-1} \tag{14}$$

where $\sum_{k+1}^{ZZ}$ is the covariance of the model predicted observation ensemble obtained from $H_k x_{k+1}^{i-}$. $\sum_{k+1}^{XZ}$ is the cross variance of the model forecast and observation prediction. After that, we resample the analyzed model state back into original layers at each HRU. The update retention parameters and soil moisture routing prior to the next step (t + 1) are calculated as the Equations (2) and (9), respectively.

Figure 2 presents the flowchart of this study with detailed steps for each of the simulation scenarios: the open-loop model (hereafter OL); the assimilation of SM9 into the SWAT model with the EnKF technique (hereafter EnKF-SM9); and the assimilation of SM1 into the SWAT model with the EnKF technique (hereafter EnKF-SM1). The DA evaluation is in the period of 2017–2019 because this is the same as the validation period of the deterministic SWAT model.

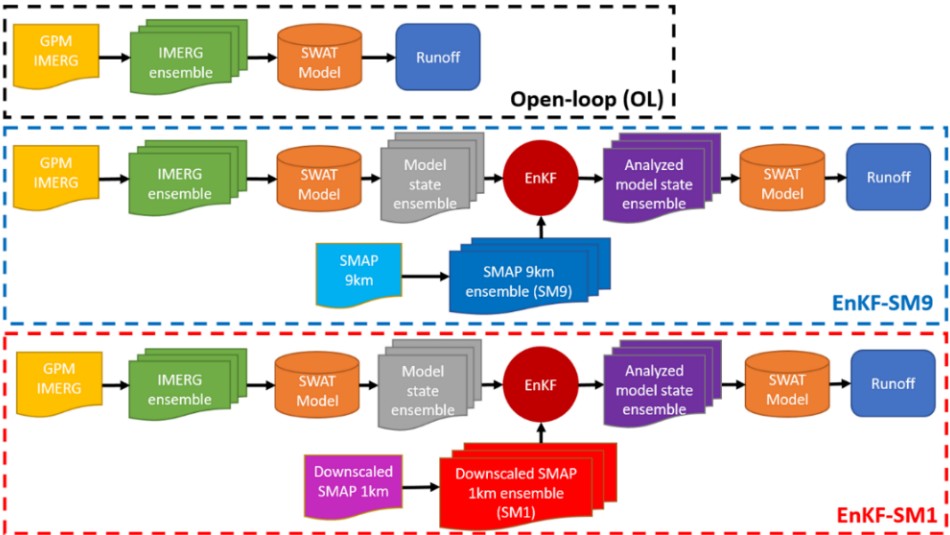

**Figure 2.** Flow chart of this study. EnKF-SM9 and EnKF-SM1 stand for streamflow simulations using the SWAT model with the state variable of SM9 and EnKF technique, and streamflow simulations using the SWAT model with the state variable of downscaled SM1 and EnKF technique, respectively.

### 3.4. Streamflow Performance Metrics

The modified Kling–Gupta efficiency (*KGE*, [75]) was used to evaluate streamflow simulations, with its formula as follows.

$$KGE = 1 - \sqrt{(r-1)^2 + (\beta-1)^2 + (\gamma-1)^2} \tag{15}$$

In which:

*r* is the Pearson correlation coefficient, reflecting the error in shape and timing between observed and simulated streamflow.
*β* is the bias term, evaluating the bias between observed and simulated streamflow.
*γ* is the ratio between coefficients of variation in observed and simulated streamflow, assessing the flow variability error with bias consideration.

We also calculated the benefit of the DA by using the Efficiency Index (*Eff*) [76], expressed as

$$Eff = 1 - \frac{\sum_{k=1}^{n}(Q_{da,k} - Q_{obs,k})^2}{\sum_{k=1}^{n}(Q_{ol,k} - Q_{obs,k})^2} \tag{16}$$

where *n* represents the total time steps. $Q_{da,k}$, $Q_{ol,k}$, and $Q_{obs,k}$ denote the simulated streamflow with data assimilation, simulated streamflow without data assimilation (open loop), and observed streamflow at time step *k*, respectively. $Eff > 0$ denotes an improvement in streamflow simulation after implementing the DA scheme and vice versa for $Eff \leq 0$.

To focus on different aspects of flow time series, we transformed the flow time series before calculating *KGE* or *Eff*, as follows [77].

- Normal streamflow time series (hereafter $Q_{nor}$), to have more weights on high flow.
- Square root streamflow time series (hereafter $Q_{sqr}$), to have more weights on average flow.
- Inverse streamflow time series (hereafter $Q_{inv}$), to have more weights on low flow.

It is noted that with inverse streamflow transformation, to avoid zero flow, we added 1/100 of mean observed flow before the transformation.

### 4. Results and Discussion

#### 4.1. Characteristics of Soil Moisture SMAP Products

During the period of 2017–2019, apart from July, the average available data for SM9 across the studied catchments is approximately 35% in each month (Figure 3). In July, a

significant reduction in coverage of SM9 (below 25%) was observed. This is likely due to a large gap in July 2019 (see Figure A1) because SMAP satellite was in a safe mode and did not provide the observed soil moisture information [78]. The averaged coverage of SM1 was only one third of that of SM9 (approximately 11.5% in each month) and was 5% in July. The reason for SM1's low coverage in July is similar to that of SM9 as the SM1 is the downscaled product of SM9 and therefore inherits the gap from SM9.

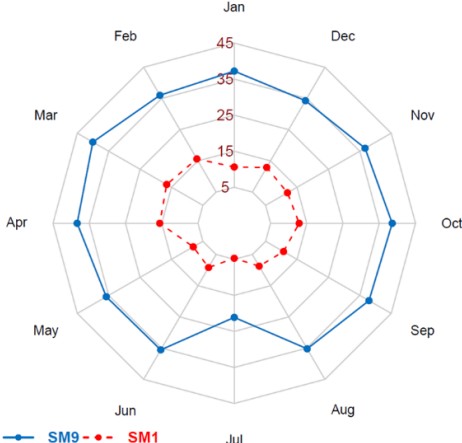

**Figure 3.** Radar chart of average soil moisture available data (in percent) over 8 catchments in each month for SMAP 9 km (SM9) and SMAP 1 km (SM1) during 2017–2019.

The relationship between estimated SM value from SM9 and SM1 presented in Figure 4. Two small catchments—gvo and aho (<500 km², Figure 4a,b)—exhibited weak correlation between the two SM datasets as compared to the larger catchments. In these small catchments, the SM1 product seems to estimate higher SM value as higher density points are observed at the lower part of 1-1 line.

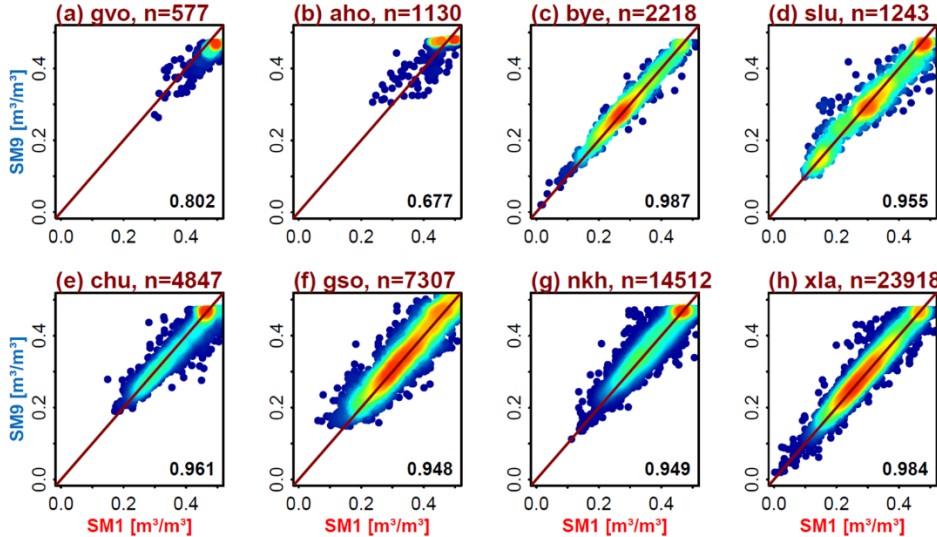

**Figure 4.** Comparison between soil moisture volume metric estimated at sub-basins over eight catchments (**a**) gvo, (**b**) aho, (**c**) bye, (**d**) slu, (**e**) chu, (**f**) gso, (**g**) nkh, and (**h**) xla using SM9 and SM1. The points colors indicate points density, with more red meaning higher points density. The values in the bottom right indicate correlation values between the two soil moisture datasets. n is the total pair days which both SM9 and SM1 have values at a sub-basin.

Figure 5 illustrates the proficiency of two SM products for reflecting a dry-down event in a medium-sized bye catchment. We used precipitation and SM to examine the drying of soil over time with respect to a rainfall event. After the rainfall event on 4 April

2018 (average 8.5 mm for the entire catchment), the catchment received less rainfall in subsequent days, and almost no rainfall after April 8. During the same period, we noted that both SM products exhibited similar dry down patterns. It is possible that SMAP observed the near-surface soil moisture conditions as they transitioned from saturated to dry conditions. Inter-comparison between these two SM products highlights the additional spatial patterns in soil moisture provided by each product. The SM1 dataset provides detailed variation in SM in space as compared to the SM9 dataset, demonstrated by its high standard deviation values (Figure 5c). However, we also see the coverage of SM1 was not complete for the entire catchment. This is because of the limited coverage of this product due to its dependence on LST data, which is influenced by cloud cover.

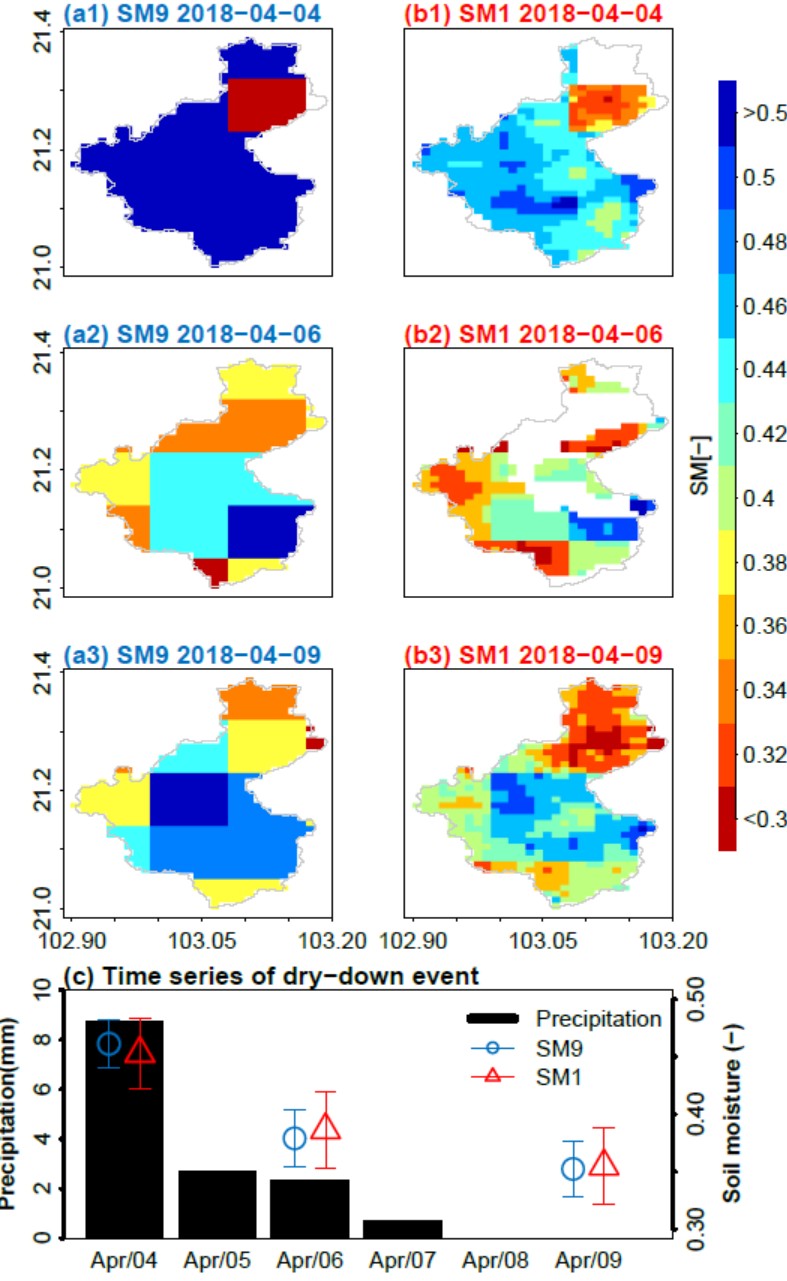

**Figure 5.** Spatial variation in a dry-down event in bye catchment from April 4, 2018, to April 9, 2018, with soil moisture SMAP 9 km (SM9, (**a1,a2,a3**)), soil moisture SMAP 1 km (SM1, (**b1,b2,b3**)), and (**c**) time series of dry-down event at the same period from GPM IMERG (black bar) and SM9 (blue) and SM1 (red). The error bars indicate standard deviation of SM variation in the catchment.

### 4.2. Performances of Deterministic Hydrological SWAT Model in Simulating Streamflow

The statistical metrics for the SWAT model are presented in Table 4, and optimized parameter sets of the SWAT model for each basin are provided in Supplementary Table S1. The model performances for high flow ($Q_{nor}$) and average flow ($Q_{sqr}$) were satisfactory, with median KGE values of calibration/validation of 0.617/0.607 for high flow and 0.702/0.695 for average flow (Table 4). The SWAT streamflow simulations are robust across the catchments (all KGE values were greater than 0.5), except for aho and slu catchments. It is likely that the rainfall patterns in these basins could be affected by topography [43,79]. The streamflow simulation for low flow ($Q_{inv}$) was relatively poor, with a median KGE of −0.263 and −0.086 for the calibration and validation periods, respectively. This poor performance for low flow has also been observed in previous studies [38].

**Table 4.** Statistical metrics for calibration and validation period with deterministic SWAT model. $KGE_{nor}$, $KGE_{sqr}$, and $KGE_{inv}$ indicate performances with $Q_{nor}$ (more weight on high flow), $Q_{sqr}$ (more weight on average flow), and $Q_{inv}$ (more weight on low flow), respectively.

| Station Name | Calibration (2013–16) | | | Validation (2017–19) | | |
|---|---|---|---|---|---|---|
| | KGE_nor | KGE_sqr | KGE_inv | KGE_nor | KGE_sqr | KGE_inv |
| gvo | 0.623 | 0.703 | 0.413 | 0.670 | 0.686 | 0.674 |
| aho | 0.486 | 0.613 | −0.984 | 0.417 | 0.462 | −0.382 |
| bye | 0.786 | 0.864 | 0.176 | 0.575 | 0.796 | 0.259 |
| slu | 0.334 | 0.598 | 0.419 | 0.303 | 0.410 | −0.089 |
| chu | 0.611 | 0.312 | −2.708 | 0.694 | 0.470 | −1.774 |
| gso | 0.757 | 0.718 | −2.727 | 0.639 | 0.704 | −0.977 |
| nkh | 0.542 | 0.700 | −0.701 | 0.513 | 0.788 | −0.082 |
| xla | 0.698 | 0.786 | 0.479 | 0.681 | 0.750 | 0.650 |
| median | 0.617 | 0.702 | −0.263 | 0.607 | 0.695 | −0.086 |

### 4.3. Temporal Variation for Open Loop, EnKF-SM9, and EnKF-SM1

Generally, soil moisture profiles across sub-basins in each catchment are mostly similar. For an illustrated purpose, we present here profiles of a sub-basin at xla river basin (>6000 km²) in terms of precipitation, estimated SM from the open loop, EnKF-SM9, and EnKF-SM1 models for topsoil layer (0–50 mm), during the year of 2019 (Figure 6). It is interesting that variation in topsoil SM does not exhibit strong correlation with variation in precipitation. This observation is different from another study in the tropical regions [16]. The relationship between topsoil SM and precipitation is even weaker when we examine it at smaller catchments (data not shown). Looking at details for typical 10-day periods in January 2019 (box A) and September 2019 (box B), we found the impacts of the DA framework on the SM simulations. Specifically, the SM simulations with the DA had drier down or more fluctuation as compared to simulations without DA, according to the variation in observed SM from SM9 and SM1. With respect to temporal simulated streamflow, the OL-based SWAT model produced results quite similar to the simulated time series from the deterministic SWAT model (Figure 7a). On the other hand, the simulated streamflow from EnKF-SM9-SWAT and EnKF-SM1-SWAT are slightly better, with higher $KGE_{sqr}$ values (Figure 7a). When we examined the error density between the observed and simulated streamflow from different simulation scenarios, the error density from EnKF-SM1-SWAT had the peak closest to the zero-error vertical line (Figure 7b).

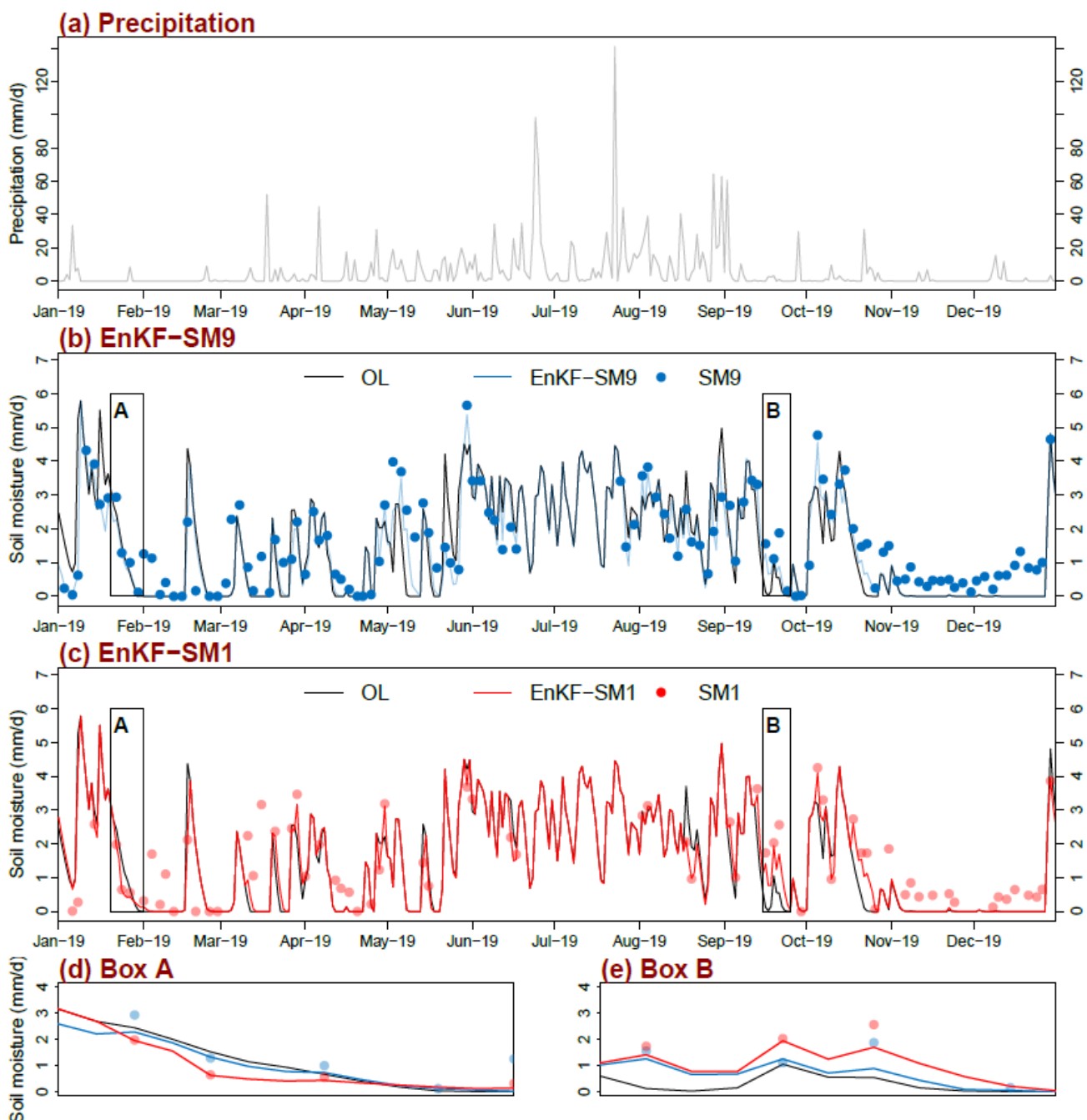

**Figure 6.** Profile of a sub-basin of xla river basin during the year of 2019 for temporal variation in (**a**) areal precipitation; (**b**) soil moisture at the topsoil layer (0–5 mm) of OL, EnKF-SM9 model and observed SM9; (**c**) soil moisture at the topsoil layer (0–50 mm) of OL, EnKF-SM1 model and observed SM1; (**d**) zoom of the last ten days in January 2019 (box A); (**e**) zoom of the last ten days in September 2019 (box B).

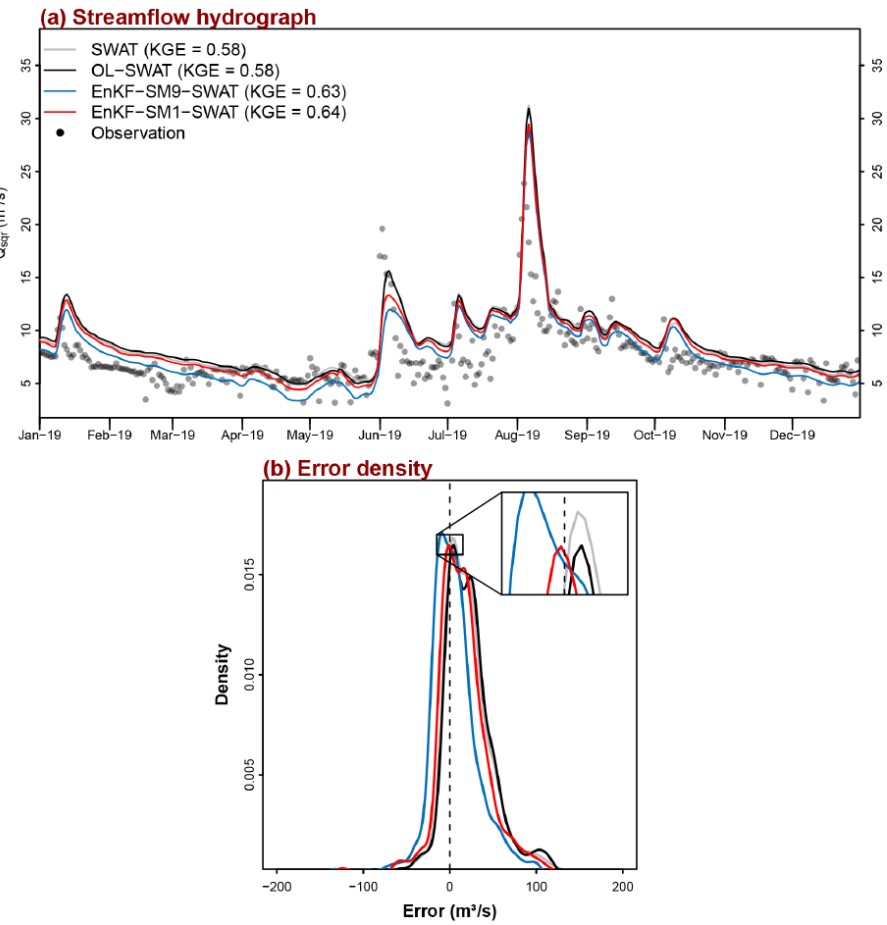

**Figure 7.** (**a**) Streamflow hydrograph comparison, and (**b**) error density between observed and simulated streamflow from different hydrological SWAT simulation scenarios during the year of 2019 at xla river basin. The black dash line in (**b**) is the zero error vertical line. The inlet panel in (**b**) zooms in the peak error density from different simulation scenarios.

### 4.4. Statistical Performances for Data Assimilation with SM9 and SM1

Figure 8 represents boxplots of streamflow simulations from the OL, EnKF-SM9, EnKF-SM1 models in two cases- all catchments (*n* = 8) and catchments >500 km$^2$ (*n* = 6). The defined error values for each basin for EnKF-SM9 and EnKF-SM1 are provided in Supplementary Tables S2 and S3, respectively. Overall, in the high flow assessment metric (Figure 8a), the EnKF-SM1 model was slightly better than the OL model at either consideration of all catchments or catchments greater than 500 km$^2$. Meanwhile, the EnKF-SM9 model was only better than the OL model in the case of catchments greater than 500 km$^2$. We interpret this result as evidence that the high-spatial SM1 is robust in all types of catchments, while the SM9 is too-coarse for small watersheds. Furthermore, the assessment of average flow provided the same conclusion (Figure 8b). This finding is similar to Abbaszadeh et al., 2020 [13], as it implies the importance of spatial resolution over temporal resolution, but is in contrast to the work of Azimi et al., 2020 [36].

On the other hand, low flow assessment (Figure 8c) revealed that the EnKF-SM9 model had a higher median KGE score than the OL-model, either at all catchments or at catchments >500 km$^2$. This may be because the OL model considers forecast error by perturbing rainfall forcing only, while the EnKF-SM9 model considers both forecast error and model error by perturbing rainfall forcing and soil moisture. The soil water content changes are more sensitive with changes in low flow in dry conditions than high flow in wet conditions or average flow.

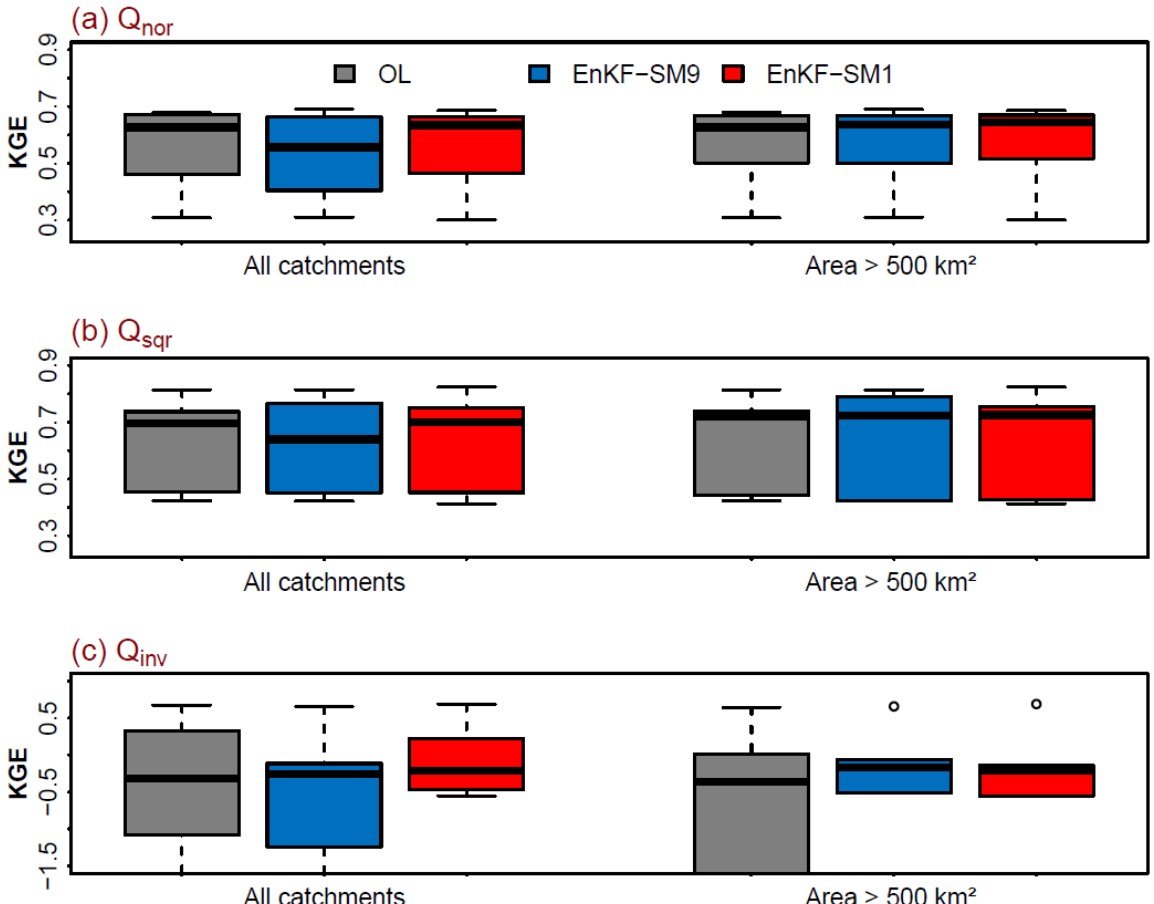

**Figure 8.** Performance metrics in streamflow simulation in (**a**) normal-, (**b**) square root-, and (**c**) inverse-time series for open loop (OL)-, EnKF-SM9-, and EnKF-SM1-based SWAT model during the period 2017–2019. With respect to all catchments, total simulated catchments are 8. With respect to catchments having an area greater than 500 km$^2$, total simulated catchments are 6.

*4.5. Assessment of Factors Impact on DA Performances*

We examined the relationship between the Efficiency index ($Eff$) with the available SM for two DA models, EnKF-SM9 and EnKF-SM1 (Figure 9). From all flow types (high, average, and low flow), the EnKF-SM1 models exhibited higher Eff scores than the EnKF-SM9 models. When we excluded small catchments (<500 km$^2$), higher Eff scores were observed for EnKF-SM models. Since SM1 has a shorter data record, our results suggest that spatial information plays a more important role than temporal information. We also found that the SM1 available day has a significant positive correlation with $Eff$ scores, while this relationship for available SM9 is not significant (see Figure A2), suggesting a potential approach for improving the high-spatial SM-based DA model that increases its temporal information.

The relationships between the $Eff$ and normalized different vegetation index (NDVI) for average flow, high flow, and low flow are given in Figure 10a–c. Catchments with dense vegetation (higher NDVI values) seem to have lower $Eff$ scores, reflecting the limitations of satellite-based SM to accurately capture soil water content at these dense vegetated catchments. This result is consistent with that of Azimi et al., 2020 [36]. However, our results provide new insight. When we compared the two SM-based models, the EnKF-SM1 seems to have less dependence with NDVI, demonstrated by its $Eff$ not being significantly reduced when NDVI values were high, as compared to the departure of $Eff$ of the EnKF-SM9 model.

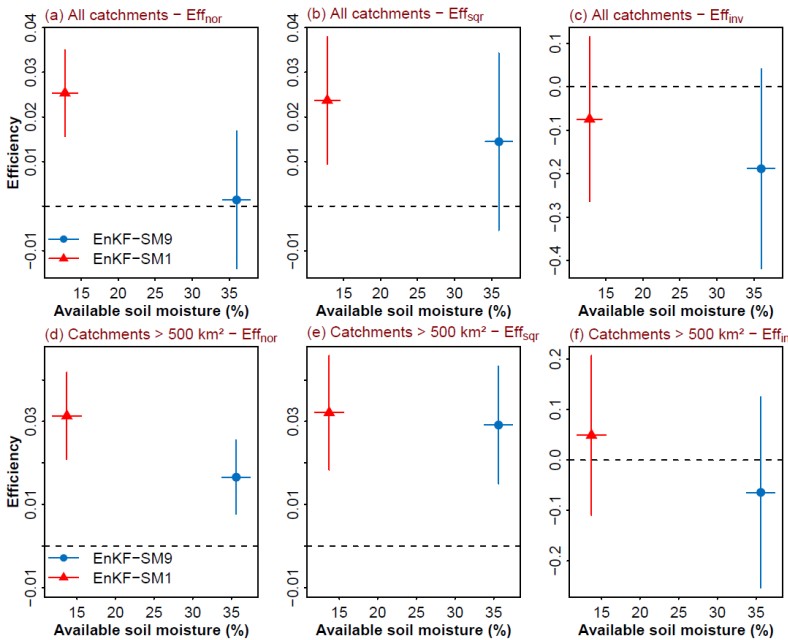

**Figure 9.** Comparison between average efficiency index of streamflow simulation using assimilation of EnKF-SM9 model and assimilation of EnKF-SM1 model and OL-based model for all catchments (**a**–**c**) and catchments > 500 km² (**d**–**f**). Points above zero-dash line indicate an improvement in streamflow simulation after implementing the data assimilation framework as compared with the OL-based model simulation.

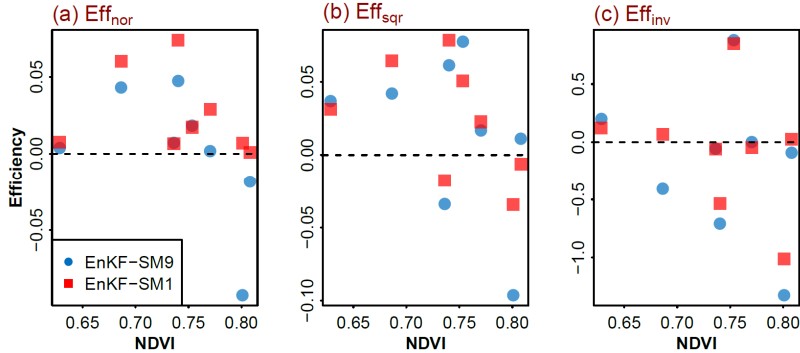

**Figure 10.** Relationship between efficiency of data assimilation for (**a**) $Eff_{nor}$(high flow score); (**b**) $Eff_{sqr}$(average flow score); and (**c**) $Eff_{inv}$(low flow score) time series with average NDVI values over eight catchments.

## 5. Conclusions and Further Study

As satellite-based remote sensing technology continues to advance, operational applications of satellite-based soil moisture products are becoming more routine. These valuable earth observations are proving to be a significant addition to several water resource management applications. However, there remain many unanswered questions regarding the most effective approach for integrating these data, as well as how temporal resolution, spatial resolution, and data record length affect their utility. The primary goal of this study was to address some of these questions and examine the trade-offs between optimal spatial vs optimal temporal resolution for two remotely sensed soil moisture (SM) products in a hydrologic data assimilation framework. Two remotely sensed SM datasets—downscaled SMAP 1 km (SM1) and SMAP 9 km (SM9)—were assimilated in the hydrological model (Soil and Water Assessment Tool, SWAT) using the Ensemble Kalman Filter (EnKF) algorithm. The effect of basin size was assessed by comparing simulated streamflow performance in eight catchments ranging in size from 267 km² to 6430 km² across tropical Vietnam.

Model fidelity was influenced by both temporal and spatial resolution, however, the DA-based models were slightly better than the open-loop models in three aspects of flow assessment with KGE metrics (low, average, and high flow). In addition, the EnKF-SM1 model was more pronounced, especially for small catchments. This indicates that the improvement in the streamflow simulation due to assimilated soil moisture is more significant in catchments where downscaled SMAP 1 km has fewer missing observations. We also found that the vegetation effects on soil moisture are less significant in the EnKF-SM1 models compared to EnKF-SM9 models, further demonstrating the reduced uncertainty in streamflow from applying the finer spatial resolution soil moisture product. To this end, this study demonstrates the potential benefits of higher spatial resolution remotely sensed SM for improving hydrologic applications.

Overall, the results of this study provide useful information for developers of satellite-based SM product for improving their soil moisture retrieval algorithms at a global scale, especially in tropical regions. In addition, we conclude that optimal strategies for the integration of satellite-based soil moisture in hydrologic models must carefully consider basin size, climate, land cover, and, perhaps most importantly, the spatial and temporal resolution of the satellite-based products.

**Supplementary Materials:** The following supporting information can be downloaded at: https://www.mdpi.com/article/10.3390/rs14071607/s1. Figure S1. Publications (peer-reviewed articles) per year related to topic of soil moisture data assimilation in hydrological model. Figure S2. The most currently studies on soil moisture data assimilation in hydrology using remotely sensed soil moisture as observed soil moisture in updating the model state variable. Table S1. Description of optimized SWAT model parameters for each basin. Table S2. Description of best guess error defined values for EnKF-SM9 model for each basin. Table S3. Description of best guess error defined values for EnKF-SM1 model for each basin.

**Author Contributions:** Conceptualization, methodology, software, visualization, validation, writing—original draft preparation, M.-H.L.; methodology, validation, software, writing—review and editing, B.Q.N.; conceptualization, methodology, writing—review and editing, H.T.P.; methodology, software, writing—review and editing, A.P.; data curation, visualization, writing—review and editing, H.X.D.; writing—review and editing, R.R.; writing—review and editing—J.D.B.; funding acquisition, writing—review and editing, supervision, V.L. All authors have read and agreed to the published version of the manuscript.

**Funding:** This research received no external funding.

**Institutional Review Board Statement:** Not applicable.

**Informed Consent Statement:** Not applicable.

**Data Availability Statement:** Not applicable.

**Acknowledgments:** We would like to express our sincere gratitude to the Vietnam Meteorological and Hydrological Administration for providing us with the required hydrological measurements that enabled this study. Special acknowledgement also goes for following institutions for providing easy access to their products (institution | products), including NASA | GPM IMERG precipitation, SMAP soil moisture, MODIS land cover, MODIS NDVI; NCAR | CFSR air temperature; FAO | HWSD soil properties; and Hydro SHEDS | DEM. We would also like to thank Bin Fang (the University of Virginia) for sharing the global downscaled SMAP 1 km from his research. We also publish codes relevant to this study, including (i) R language codes for preparation of forcing inputs (climatic data and catchment attributes) for multiple swat projects (version 2012)" in the following link https://github.com/mhle510/swatIP, accessed on 28 January 2022; (ii) R language codes and complied execution file for SMAP data assimilation for hydrologic SWAT model streamflow simulation using Ensemble Kalman Filter in the following link https://github.com/mhle510/smap_enkf_swat, accessed on 28 January 2022; and (iii) Fortran source codes for EnKF algorithms with SWAT version 2012 in the following link https://github.com/amolpatil771/SWAT_DA, accessed on 28 January 2022.

**Conflicts of Interest:** The authors declare no conflict of interest.

## Appendix A

**Table A1.** Characteristics of climatic conditions and catchment attributes in eight studied catchments. The precipitation and potential evapotranspiration in each catchment are estimated from the calibrated SWAT model for the entire area of that catchment.

| Types | Data Description | Spatial Resolution | gvo Benhai River | aho Trakhuc River | bye Namnua River | slu Luy River | chu LucNam River | gso Krong Ana River | nkh Hieu River | xla Ma River |
|---|---|---|---|---|---|---|---|---|---|---|
| Area (km$^2$) | | | 267 | 383 | 638 | 964 | 2090 | 3100 | 4024 | 6430 |
| Dry Season/Wet Season | | | I–VIII/IX–XII | I–VIII/IX–XII | XI–IV/V–X | XI–IV/V–X | XI–IV/V–X | XII–IV/V–XI | XII–V/VI–XI | XI–IV/V–X |
| Precipitation (unit in mm) | IMERG Final v6 | ~10 km | 1911 | 2165 | 1644 | 1577 | 1807 | 1798 | 1755 | 1629 |
| Potential Evapotranspiration (unit in mm) | Hargreaves method with data from CFSR vs2 | ~25 km | 1024 | 849 | 1051 | 788 | 1258 | 1223 | 1018 | 1402 |
| Digital Elevation (DEM) (unit in m) | HydroSHEDs | 90 m | Min: 10 Max: 1213 Mean: 215 | Min: 19 Max: 1008 Mean: 366 | Min: 470 Max: 1736 Mean: 945 | Min: 25 Max: 1747 Mean: 451 | Min: 7 Max: 1003 Mean: 248 | Min: 407 Max: 2407 Mean: 658 | Min: 33 Max: 2416 Mean: 396 | Min: 282 Max: 2164 Mean: 958 |
| Land use * | MODIS12Q1 | 500 m | FRSE (50.36) SHRB (47.18) | FRSE (67.10) SHRB (31.31) | FRSE (32.07) SHRB (63.75) | FRSE (46.15) CRGR (18.02) SHRB (16.97) FRSD (11.5) | SHRB (70.67) FRSE (27.84) | CRGR (41.10) SHRB (30.04) FRSE (26.51) | SHRB (45.94) FRSE (42.85) | SHRB (75.97) FRSE (18.44) |
| Soil ** | HWSD | 1 km | Ao (100) | Ao (98.67) | Ao (100) | Ao (77.26) Lc (18.64) | Ao (92.95) Af (5.58) | Fr (39.62) Af (30.21) Ao (30.09) | Ao (98.85) | Ao (100) |
| Sub-basins, HRUs | 10% soil, 10% land use, 10% slope | | 5 sub-basins 24 HRUs | 9 sub-basins 50 HRUs | 9 sub-basins 60 HRUs | 17 sub-basins 116 HRUs | 35 sub-basins 186 HRUs | 59 sub-basins 314 HRUs | 91 sub-basins 590 HRUs | 125 sub-basins 579 HRUs |

Note: * Full name for land use- 'FRSE' Evergreen forests, 'FRSD' Deciduous forests, 'SHRB' shrubland, 'CRGR' cropland. Only major land use (>5% of total catchment area) or the first four major land use are listed. Values in blanket are percentage value over total catchment area. ** Full name for soil data- 'Ao' Orthic Acrisols, 'Af' Ferric Acrisols, 'Fr' Rhodic Ferralsols, 'Lc' Chromic Luvisol. Only major soil (>5% of total catchment area) or the first four major soil are listed. Values in blanket are percentage value over total catchment area.

**Table A2.** Name, description, range and control processes of SWAT parameters. "r_", "v_", and "a_" refer to modify the default value by making a relative change to the default value, replacing the default value by the specific value and adding a specific value, respectively.

| Parameter Name | Units | Description | Default | Range | Process |
|---|---|---|---|---|---|
| R_CN2.mgt | none | SCS runoff curve number | HRU specific | −0.25, +0.25 | Surface Runoff |
| V_SURLAG.bsn | none | Surface runoff lag time | 4 | 0.05, +24 | Surface Runoff |
| R_HRU_SLP.hru | m/m | Average slope steepness | 0.217 | −0.25, +0.25 | Surface Runoff |
| V_GW_REVAP.gw | none | Groundwater "revap" coefficient | 0.02 | 0.02, +2 | Evapotranspiration |
| V_ESCO.hru | none | Soil evaporation compensation factor | 0.95 | 0, +1 | Evapotranspiration |
| V_CH_N2.rte | none | Manning's "n" value for the main channel | 0.014 | 0, +0.3 | Channel |
| V_CH_K2.rte | mm/hour | Effective hydraulic conductivity in main channel alluvium | 0 | 0, +500 | Channel |
| R_SOL_AWC(..).sol | mm $H_2O$/ mm soil | Available water capacity of the soil layer | 0.1112 | −0.25, +0.25 | Soil |
| R_SOL_K(..).sol | mm/hour | Saturated hydraulic conductivity | 7.113 | −0.25, +0.25 | Soil |
| V_ALPHA_BF.gw | days | Base flow alpha factor | 0.048 | 0, +1 | Groundwater |
| V_GW_DELAY.gw | days | Groundwater delay | 31 | 0, +500 | Groundwater |
| V_GWQMN.gw | mm $H_2O$ | Threshold depth of water in the shallow aquifer required for return flow to occur | 1000 | 0, +5000 | Groundwater |
| V_RCHRG_DP.gw | None | Deep aquifer percolation fraction | 0.05 | 0, +1 | Groundwater |

**Table A3.** Name, description and the range of perturbation defined errors of the EnKF data assimilation framework.

| Perturbation Variables | Description | Range |
|---|---|---|
| Observed soil moisture | Observed soil moisture coefficient | 50–200 |
| Precipitation | Precipitation error coefficient | 0.1–1.0 |
| Field capacity for soil layer 1 | Field capacity for soil layer 1 coefficient | 0.1-0.3 |
| Field capacity for soil layer 2 | Field capacity for soil layer 2 coefficient | 0.05–0.2 |
| Field capacity for soil layer 3 | Field capacity for soil layer 3 coefficient | 0.01–0.1 |
| Soil moisture layer 1 | Soil moisture error standard deviation for layer 1 | 0.01–0.1 |
| Soil moisture layer 2 | Soil moisture error standard deviation for layer 2 | 0.01–0.1 |
| Soil moisture layer 3 | Soil moisture error standard deviation for layer 3 | 0.01–0.1 |
| Curve number | Curve number error standard | 1–5 |

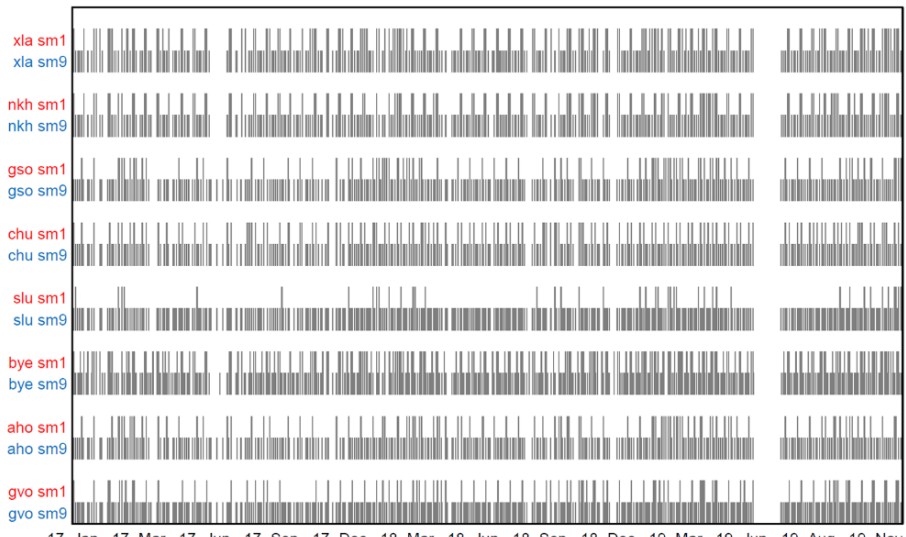

**Figure A1.** Available soil moisture (grey rectangular) for SMAP 9 km (SM9) and downscaled SMAP 1 km (SM1) at each catchment during 2017–2019. The *y*-axis label is written as hydrological station name and soil moisture products. An available soil moisture day is counted as at least 30% of basin area has soil moisture pixels.

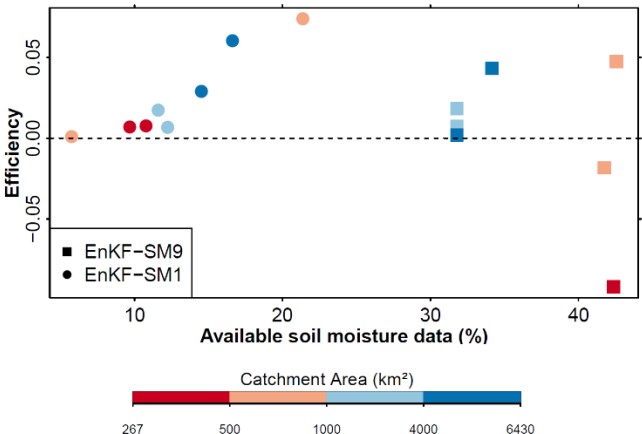

**Figure A2.** Relationship between the efficiency index and available soil moisture with the $Q_{nor}$ time series.

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
