# Peer review of "Assimilation of SMAP Products for Improving Streamflow Simulations over Tropical Climate Region—Is Spatial Information More Important Than Temporal Information?"

_remotesensing, doi:10.3390/rs14071607_

Round 1
Reviewer 1 Report
The submitted paper by Le et al. titled: "Assimilation of SMAP products for improving streamflow simulations over tropical climate region - Is spatial information more important than temporal information?", deals with the improvement of Soil and Water Assessment Tool (SWAT) model focused on streamflow simulations. The authors examined two soil moisture products in order to analyze the differences of 3 years simulation in difference basins in tropical environment. The methodology and steps are analyzed clearly. The sources of the data which are used, and their characteristics are also referred. The subject is interesting as the use of satellite soil moisture data for hydrological research is under-progress research. The paper is almost ready for publication.
My recommendation is moderate revisions.
General Comments
-Change keywords. Avoid using the same words that already exist in the title.
-Improve the abstract. Try to gain the reader’s interest.
-Figure 1. I don’t understand the importance of this figure. I know that the basins are located in different elevation and distance from the sea. However, the current figure does not provide extra details about the study area. Try another figure.
-Why these basins are so important for simulation analysis? Provide the importance of SWAT model based on the conditions and the available data.
-Provide more information about the study area (morphology, hydrology, hydrogeology etc).
-Conclusions should provide only the take home message.
-Did you used ArcGIS or QGIS?
Author Response
Please find our response in the attachment file.

Reviewer 2 Report
I like a subject of your paper and a way you present data.
Author Response
Thank you for your time to review our manuscripts
Reviewer 3 Report
General comments:
The authors explored data assimilation in SWAT model over eight catchments across tropical Vietnam with two soil moisture datasets, with several findings. The authors claimed that the research over tropical regions--- an under-represented in the literature, but the manuscript confused me a lot about the contribution or new findings (or not). I recommend a Major Revision.
Firstly, the introduction should acknowledge the contributions from the literature in great detail, this study explored data assimilation in SWAT model over tropical regions, that’s quite under-represented in the literature. But, previous studies on assimilating hydrological models with SM datasets are missing in the introduction. For example, 1. “Assimilation of Sentinel 1 and SMAP – based satellite soil moisture retrievals into SWAT hydrological model: the impact of satellite revisit time and product spatial resolution on flood simulations in small basins”. 2020 JOH; 2. High spatial resolution simulation of profile soil moisture by assimilating multi-source remote-sensed information into a distributed hydrological model, 2021 JOH, directly from Google search. At this version of introduction, the authors simply presented two research pathways, “trade-off between temporal and spatial time scale of SM products” and “large samples of catchments should be conducted”, how rough! We should not oversell our stories. So, please add more details and give readers an overview of data assimilation in hydrological model, and your real contribution.
Secondly, without discussions, we could not see any new findings of this study. Why tropical regions? any new findings or differences compared with previous studies? I could not find any discussion and new findings.
Line 370, not clear about the “local”
Line 378, why this sub-basin?
Figure 5, why these three days? Obviously, some systematic errors can be found in (a3) and (b3).
Figure 7, should merge four subplots into one, so that we could see the differences.
Line 471, robustness?
Author Response

(The authors gave the same response as above.)

Round 2
Reviewer 3 Report
I would suggest an acceptance.